# Carbon Monoxide Alleviates Salt-Induced Oxidative Damage in *Sorghum bicolor* by Inducing the Expression of Proline Biosynthesis and Antioxidant Genes

**DOI:** 10.3390/plants13060782

**Published:** 2024-03-10

**Authors:** Vivian Chigozie Ikebudu, Mulisa Nkuna, Nzumbululo Ndou, Rachel Fanelwa Ajayi, Stephen Chivasa, Katrina Cornish, Takalani Mulaudzi

**Affiliations:** 1Life Sciences Building, Department of Biotechnology, University of the Western Cape, Private Bag X17, Bellville 7535, South Africa; 3206244@myuwc.ac.za (V.C.I.); 4078438@myuwc.ac.za (M.N.); 3992677@myuwc.ac.za (N.N.); 2SensorLab, Department of Chemical Sciences, University of the Western Cape, Private Bag X17, Bellville 7535, South Africa; rngece@uwc.ac.za; 3Biosciences Department, Durham University, Durham DH1 3LE, UK; stephen.chivasa@durham.ac.uk; 4Department of Horticulture and Crop Science, Ohio Agricultural Research and Development Center, The Ohio State University, 1680 Madison Avenue, Wooster, OH 44691-4096, USA; cornish.19@osu.edu; 5Department of Food, Agriculture and Biological Engineering, Ohio Agricultural Research and Development Center, The Ohio State University, 1680 Madison Avenue, Wooster, OH 44691-4096, USA

**Keywords:** antioxidant, carbon monoxide, epidermis, hematin, heme oxygenase, salinity, oxidative stress, ROS, vascular bundles

## Abstract

Crop growth and yield are affected by salinity, which causes oxidative damage to plant cells. Plants respond to salinity by maintaining cellular osmotic balance, regulating ion transport, and enhancing the expression of stress-responsive genes, thereby inducing tolerance. As a byproduct of heme oxygenase (HO)-mediated degradation of heme, carbon monoxide (CO) regulates plant responses to salinity. This study investigated a CO-mediated salt stress tolerance mechanism in sorghum seedlings during germination. Sorghum seeds were germinated in the presence of 250 mM NaCl only, or in combination with a CO donor (1 and 1.5 μM hematin), HO inhibitor (5 and 10 μM zinc protoporphyrin IX; ZnPPIX), and hemoglobin (0.1 g/L Hb). Salt stress decreased the germination index (47.73%) and root length (74.31%), while hydrogen peroxide (H_2_O_2_) (193.5%), and proline (475%) contents increased. This increase correlated with induced HO (137.68%) activity and transcripts of ion-exchanger and antioxidant genes. Salt stress modified vascular bundle structure, increased metaxylem pit size (42.2%) and the Na^+^/K^+^ ratio (2.06) and altered primary and secondary metabolites. However, exogenous CO (1 μM hematin) increased the germination index (63.01%) and root length (150.59%), while H_2_O_2_ (21.94%) content decreased under salt stress. Carbon monoxide further increased proline (147.62%), restored the vascular bundle structure, decreased the metaxylem pit size (31.2%) and Na^+^/K^+^ ratio (1.46), and attenuated changes observed on primary and secondary metabolites under salt stress. Carbon monoxide increased HO activity (30.49%), protein content, and antioxidant gene transcripts. The alleviatory role of CO was abolished by Hb, whereas HO activity was slightly inhibited by ZnPPIX under salt stress. These results suggest that CO elicited salt stress tolerance by reducing oxidative damage through osmotic adjustment and by regulating the expression of HO_1_ and the ion exchanger and antioxidant transcripts.

## 1. Introduction

Sorghum (*Sorghum bicolor* L.) is the fifth most important cereal crop after rice, maize, wheat, and barley [1,2,3]. It is used as a staple food for human consumption and livestock feed in Africa, South Asia, and Central America, and as raw material in the production of green fuels, including bioethanol and biogas [4]. Sorghum is a model C_4_ photosynthetic plant with high biomass, a diploid genome of 750 bp, and germplasm genetic diversity to improve agronomic traits [5]. It is well-adapted to semi-arid and arid regions due to its drought and salinity tolerance traits. It adapts by maintaining ion homeostasis, activating antioxidant enzymes, osmotic regulatory systems, and cell detoxification [6,7,8]. The germination and early developmental stages are best used to evaluate the effects of salinity [9], with differential effects of genotype and level of salinity stress applied [10]. Sorghum germination and development were significantly reduced under varying salt stress levels [6,11,12,13,14]. Exposure to high salt concentrations can result in reduced agricultural productivity [10,15] and lead to food insecurity, so understanding how Sorghum tolerates salt stress is very important.

Salinity is the accumulation of salt in the soil above 40 mM NaCl and an osmotic pressure of 0.2 MPa [16]. It is one of the main abiotic stresses affecting plant development and productivity [17]. Salinity can be classified as the primary stress caused by natural occurrences such as weathering and rain, or secondary salinity caused by activities such as land clearing, deforestation, and irrigation [18]. It is estimated that 20% of all cultivated land and 33% of irrigated land are already salinized, and more than 50% will be affected by 2050 [19]. Estimates have shown that over 50% of crop yield losses worldwide result from abiotic stress, and severe stress leads to an increase in the annual loss of arable land [17,18,19]. The accumulation of high levels of salt (NaCl, NaSO_4_, Na_2_CO_3_, and NaHCO_3_) inhibits seed germination by preventing water uptake because of the low osmotic potential and specific ion toxicity (Na^+^ and Cl^−^) of seed embryos [20]. Salt stress reduces the germination percentage, germination speed, shoot and root length of canola [21], basil [22] and sorghum [12]. High-salinity stress reduces physiological processes linked with decreased stomatal conductance, which affects photosynthesis, chlorophyll content, and transpiration [23,24]. Salinity induces osmotic and ionic imbalance-mediated oxidative damage, which leads to lipid peroxidation triggered by overproduction of reactive oxygen species (ROS) such as hydrogen peroxide (H_2_O_2_), superoxide radicals (O_2_^−^), hydroxyl radicals (OH), and singlet oxygen (^1^O_2_) [25,26]. Increased levels of ROS cause toxicity to the cell and disturb redox homeostasis, which hinders cell division and plant growth [18,27]. Oxidative stress caused by increased ROS also results in nutrient imbalance, membrane damage, inhibition of enzymatic activities, and disruption of many physiological and biochemical plant growth-promoting processes, eventually leading to plant death [28,29,30]. Plants adapt to salinity by modulating osmotic adjustment and ion homeostasis, and by inducing antioxidant mechanisms that scavenge and detoxify ROS [15,31]. Osmolytes, including soluble sugars, glycine betaine, and proline compounds, accumulate in some plants to maintain water uptake and metabolic activity, thereby improving stress tolerance [16,32,33,34,35].

Improving the efficiency of ionic and osmotic homeostasis, and capacity of antioxidant systems, is required to develop salt stress-tolerant crops [29]. Antioxidant enzymes scavenge ROS under salinity stress [15,31] protecting plants against cellular damage and lipid peroxidation. These enzymes include superoxide dismutase (SOD), catalase (CAT), glutathione reductase (GR), guaiacol peroxidase (GPOX), and ascorbate peroxidase (APX) [15,31]. Heme oxygenase (HO) is an antioxidant stress-responsive protein that catalyzes the degradation of heme to form biliverdin-IXα (BV), carbon monoxide (CO), and free iron (Fe^2+^) [36]. The heme oxygenase family consists of (i) the HO1 subfamily, which contains the HO1, HO3, and HO4 proteins, and (ii) the single-member HO2 subfamily with only the HO2 protein [36,37].

Carbon monoxide, as a by-product of heme degradation, plays an important biological role in both animals and plants. In animals, it modulates anti-proliferative, anti-inflammatory, and cytoprotective signaling processes [38,39,40]. However, the full range of CO functions in plants is only beginning to emerge. CO is a signaling molecule in major physiological processes [41] and in plant growth and development [42]. Also, CO promotes lateral root formation [43], stomatal closure [44], and delays gibberellin (GA)-induced programmed cell death [45]. Heme oxygenase is the main enzymatic source of CO in plants through enzymatic degradation of heme to biliverdin IXα (BV), which also releases iron (Fe^2+^). Exogenously applied low levels of CO alter plant responses to salt stress. CO ameliorates the inhibitory effects of salinity on *Cassia obtusifolia* L. seed germination [46] and oxidative stress in rice and wheat seedlings [47,48,49,50]. Several studies have reported on the effective use of exogenously applied phyto-protectants such as chitosan [12], calcium ion [6], methyl jasmonate [11], and molybdenum [13] in mitigating the effect of salt stress in sorghum. The role of CO in ameliorating the effects of abiotic stress in sorghum remains elusive.

In this study, we investigated the role of exogenously applied CO, in alleviating salt-induced oxidative stress and improving the tolerance of sorghum by modifying its ion homeostasis and antioxidant systems.

## 2. Results

### 2.1. Carbon Monoxide Improves Seed Germination and Root Growth of Sorghum under Salt Stress

Salt negatively affected sorghum germination and reduced the germination index (GI) by 47.73% and root length by 74.31% (Figure 1A,C). However, cotreatment with CO significantly attenuated the salt-induced decrease in GI, with no statistically significant difference between the control GI and cotreatments of salt with the two levels of CO (Figure 1A). While cotreatment with CO did not fully protect root growth from salt stress, there was a significant partial rescue diminishing the reduction in root length from 74.31% down to 35.6% (1 μM CO) and 52.98% (1.5 μM CO). However, when compared to the sorghum seedlings treated with salt only, cotreatment increased root lengths by 150.59% (1 μM CO) and 83.2% (1.5 μM CO), (Figure 1C). A CO scavenger (hemoglobin) and an HO1 inhibitor (ZnPPIX) [51] were exogenously applied to CO-treated sorghum seedlings under salt stress to hinder the alleviatory role of CO. While exogenous application of hemoglobin (0.1 g/L) did not significantly block the salt effects on the GI (Figure 1B), it significantly reduced root length by 43.5% (Figure 1D) in seedlings treated with 1 μM CO. However, ZnPPIX treatment had no significant effects on the growth of sorghum seedlings.

### 2.2. CO Reduces Salt-Induced Oxidative Damage in Sorghum Seedlings

The role of CO in the alleviation of salt-induced oxidative damage was investigated based on the levels of ROS by monitoring H_2_O_2_ content (Figure 2). Production and distribution of H_2_O_2_ induced by salt stress were assessed by DAB histochemical staining, which produces a deep brown DAB and H_2_O_2_ reaction product (Figure 2A). Exogenous application of 1 μM CO to 250 mM NaCl (salt)-treated sorghum seedlings alleviated the salt-induced oxidative stress (lighter brown color indistinguishable from that of the control samples). However, treatment with hemoglobin and ZnPPIX countered the alleviatory role of CO and oxidative damage was unimpeded, as reflected by the dark brown root staining (Figure 2A).

Salt stress significantly increased seedling H_2_O_2_ (by 193.5%) compared to the control seedlings (Figure 2B), indicating oxidative damage to the lipid membranes. Exogenous 1 μM CO reduced H_2_O_2_ accumulation in salt-stressed sorghum seedlings. Similarly, ZnPPIX reduced the ameliorating role of CO, resulting in an increased H_2_O_2_ content by 29.23% (in 5 μM ZnPPIX) and 38.78% (in 10 μM ZnPPIX) in the sorghum seedlings (Figure 2B) and oxidative damage was apparent.

### 2.3. CO Increases Proline and Total Soluble Sugar Content in Sorghum under Salt Stress

The effects of CO on the osmoregulation of salt-stressed sorghum were assessed by monitoring the accumulation of proline and changes in the transcript levels of *Sorghum bicolor PYRROLINE-5-CARBOXYLATE SYNTHETASE 1* (*SbP5CS1*), a gene involved in the proline biosynthesis pathway (Figure 3). Salt stress increased proline content by 475%, (Figure 3A). Proline accumulation further increased in response to exogenous application of CO by 1147.62% (1 μM CO) and 76.28% (1.5 μM CO), (Figure 3A,B). However, the stimulatory effect of CO (1.5 μM CO) on proline content in salt-stressed sorghum seedlings was repressed by the CO scavenger, resulting in a 29.3% (0.1 g/L Hb) decrease. The HO1 inhibitor also led to a decrease in proline content by 22.73% (5 μM ZnPPIX) and 18.56% (10 μM ZnPPIX) under salt stress (Figure 3B). Salt stress alone did not change the transcript level of sorghum *P5CS1* (*SbP5CS1*), (Figure 3C). However, 1 μM and 1.5 μM CO slightly increased the transcript levels of *SbP5CS1* under salt stress. Carbon monoxide was scavenged by 0.1 g/L Hb, whereas HO activity (the ability to produce CO) was inhibited by 5 μM and 10 μM of ZnPPIX shown by a decrease in *SbP5CS1* transcript under salt stress (Figure 3C).

### 2.4. Morphology and Element Analysis

#### 2.4.1. Co Improves the Morphology of Sorghum Anatomy

To further determine the effects of salt stress and the alleviatory role of CO, the morphology of the vascular bundles of sorghum, which is made up of xylem, phloem, and pits, was analyzed using scanning electron microscopy (Figure 4). The control seedlings had smooth and round shaped xylem tissue (Figure 4A). Salt stress induced deformation of vascular bundle layers, causing stretched meta xylem and phloem (Figure 4B), and the pits enlarged by 42.2% (Figure 4F,I). Treatment with CO subdued these changes (Figure 4C,D). However, although 1 μM CO significantly decreased the pit size by 31.2% compared to the salt-stressed seedlings, the higher 1.5 μM CO level caused a 17.4% additional increase in pit size (Figure 4G–I).

#### 2.4.2. CO Reduces Na^+^ Toxicity and Improves K^+^ Content

The effect of salt stress on nutrient absorption by sorghum seedlings was analyzed using scanning electron microscopy–energy dispersion X-ray spectroscopy (SEM-EDX) (Figure 5A–D; Appendix A). Salt stress resulted in a significant increase in Na^+^ content (1788.24%) and a decrease in K^+^ content (160%), showing a much larger Na^+^/K^+^ ratio (2.06) increase of 635.7% (Figure 5B) than untreated seedlings (Na^+^/K^+^ of 0.28), (Figure 5A). Interestingly, CO decreased the level of Na^+^ absorption by 12.46% (1 μM CO) and 17.45% (1.5 μM CO), whereas K^+^ absorption was significantly increased by 23.08% (1 μM CO) and 33.97% (1.5 μM CO). As a result, the Na^+^/K^+^ ratios were 29.13% and 38.35% in sorghum seedlings treated with salt and 1 μM or 1.5 μM of CO (Figure 5C,D).

To further understand the role of CO in ion homeostasis in sorghum seedlings, the transcript levels of *Sorghum bicolor vacuolar Na^+^/H^+^ exchanger antiporter (SbNHX4)* and *Sorghum bicolor potassium ion transporter* (*SbKT1)* genes were measured (Figure 5E). These genes were constitutively expressed with *SbKT1* having the highest level of expression under control conditions (Figure 5E). Salt stress increased the transcript levels of *SbNHX4* and *SbKT1*, however, under the influence of 1 μM of CO in salt-stressed seedlings, their transcript levels were downregulated. Conversely, 1.5 μM of CO increased the transcript level of *SbNHX4*, whereas *SbKT1* transcript level was decreased under salt stress. CO scavenger (0.1 g/L Hb) and 10 μM of ZnPPIX decreased transcripts while 5 μM of ZnPPIX treatment increased transcript levels of *SbNH4* and *SbKT1* in the presence of 1 μM of CO under salt stress (Figure 5E).

#### 2.4.3. CO Stabilizes the Nature and Structure of Biomolecules

Fourier-transform infrared (FTIR) spectroscopy was used to determine the biodegradation of biomolecules through the visualization of chemical bonds found in the proteins, lipids, and carbohydrates of sorghum seedlings under salt and hematin treatments. The mid-IR spectrum of 4000–400 cm^−1^ was used for this analysis (Figure 6). The spectral peak at 3304.7 cm^−1^ represents the polymeric –OH stretch vibrations under the frequency range 3700–3200 cm^−1^ indicating the presence of phenols and alcohols. The peaks at 2923.48 cm^−1^ and 1449 cm^−1^ represent the C-H stretch vibration within the frequency range of 3000–2850 cm^−1^ and 1485–1445 cm^−1^, respectively, an indication of the presence of saturated aliphatic alkanes in lipids. Peaks 2139.29 cm^−1^ and 1664.45 cm^−1^ represent the -C≡C, -C=C- stretch vibration within the frequencies 2140–2100 cm^−1^ and 1680–1620 cm^−1^, respectively, indicating the presence of olefinic alkynes and alkenes in carbohydrates. Peaks 1233, 1150, and 1025 cm^−1^ represent the -C-N stretch vibration within the frequency range of 1250–1020 cm^−1^ indicating the presence of primary, secondary, and tertiary amines (proteins) as well as silicate ions (1100–900 cm^−1^). Peak 852 cm^−1^ (890–820 cm^−1^) represents peroxide, while peaks 759, 701 (within frequencies 800–700 cm^−1^ and 715–570 cm^−1^) and 563.95 cm^−1^ (600–500 cm^−1^) represent the C-Cl/C-I aliphatic choro/Iodo halogenated compounds, C-S thioesters, and S-S disulfides. The FTIR spectra of salt-stressed sorghum seedlings showed peak shifts between 3300 and 2924 cm^−1^ (phenols and lipids), 1664.45 (carbohydrates), 1449 (lipids), 1150 (proteins), while peaks at 759, 701, and 563.95 cm^−1^ indicate halogenated compounds. However, exogenously applied hematin (CO1, blue spectra) under salt stress moved the spectral peak shifts to a pattern more similar to the control (black spectra) more than the CO2 (green spectra).

### 2.5. Effect of CO on the Activity and Expression of Heme Oxygenase in Sorghum under Salt Stress

To determine the effects of salt and the alleviatory role of CO on antioxidant systems, the activity of heme oxygenase (HO) was assayed together with establishing the response at the transcription and translation levels (Figure 7). Salt stress induced HO activity by 137.68% (Figure 7A), with combined treatment using salt and 1 μM of CO further increasing HO activity by 30.49% (Figure 7A). However, cotreatments with 0.1 g/L of Hb or 10 μM of ZnPPIX significantly inhibited the alleviatory role of CO on enzyme activity under salt stress (Figure 7A).

Western blot analysis detected a 34 kDa HO1 protein in sorghum seedlings exposed to salt stress, CO, Hb, and ZnPPIX (Figure 7B). The HO1 protein band was stronger in 1 μM of CO (Lane 3) and in 1.5 μM of CO treatments (Lane 4) under salt stress than in the salt-only treatment (Lane 2). The expression level of the HO1 protein in sorghum seedlings treated with the ZnPPIX (Lane 5) and Hb (Lane 6) treatments was less than in seedlings treated with CO and salt stress (Figure 7C). For the positive control, the anti-HSP70 monoclonal antibody was used to detect the 70 kDa HSP70 protein (Appendix A).

The transcript level of *Sorghum bicolor heme oxygenase 1* (*SbHO1*) was constitutively expressed under control conditions (Figure 7C). Salt stress decreased *SbHO1* transcription levels compared to the control. Treatment with 1 μM and 1.5 μM of CO further decreased *SbHO1* transcription under salt stress. In the presence of 1 μM of CO, the *SbHO1* transcription level was further scavenged by 0.1 g/L of Hb and the HO inhibitor, ZnPPIX (5 μM and 10 μM of ZnPPIX) under salt stress (Figure 7C).

### 2.6. Influence of CO on the Transcript Level of Antioxidant Genes in Sorghum Seedlings under Salt Stress

The transcript levels of *Sorghum bicolor Iron Superoxide dismutase* (*SbFeSOD*), *Sorghum bicolor Manganese Superoxide dismutase* (*SbMnSOD*), and *Sorghum bicolor Catalase* (*SbCAT*) were analyzed using semi-quantitative reverse transcription–PCR (RT-PCR). All the genes were constitutively expressed (Figure 8). Salt stress increased the *SbFeSOD* transcript level but decreased the *SbMnSOD* and *SbCAT* transcript levels. The transcript levels of *SbFeSOD* and *SbMnSOD*, were upregulated by 1 μM of CO, whereas the *SbCAT* transcript level was downregulated under salt stress. Additionally, a high CO concentration (1.5 μM of CO) increased the *SbFeSOD* and *SbMnSOD* transcript levels; however, the *SbCAT* transcript level decreased. The CO scavenger (0.1 g/L of Hb) and the CO inhibitors (5 μM and 10 μM of ZnPPIX), decreased the *SbFeSOD*, *SbMnSOD*, and *SbCAT* transcripts levels in the presence of 1 μM of CO and salt stress (Figure 8).

### 2.7. Pearson’s Traits Correlations

The relationship between different traits studied was analyzed using Pearson’s correlation (Figure 9). In this study, the result showed that proline, pit diameter, Na^+^/K^+^, were strongly correlated with a correlation value r closer to 1 SbHO1 enzyme activity, were also positively correlated with a correlation value of r closer to 1. Seedlings root length was found to be strongly correlated with the germination index (r = 0.9). Oxidative marker H_2_O_2_ was strongly correlated with proline (r = 0.6), pit diameter (r = 0.83), Na^+^/K^+^ (r = 0.8), and HO1 enzyme activity (r = 0.5). H_2_O_2_ exhibited negative correlations with germination index and root length with a correlation value r closer to −1. Furthermore, proline was positively correlated to Na^+^/K^+^ (r = 0.5), and SbHO1 enzyme activity (r = 0.70).

## 3. Discussion

Sorghum is a moderately salt- and drought-tolerant crop, and its adaptation is associated with maintaining ion homeostasis, activating antioxidant enzymes, deploying osmotic regulatory metabolites, and engaging cell detoxification processes [6,7,8]. However, extended exposure to abiotic stresses, such as water deficit and high soil salt concentrations, result in reduced agricultural productivity [10,15]. Soil salinity induces osmotic stress due to dissolved salt in the soil solution, which limits plant water uptake and induces ion toxicity, thereby adversely affecting seed germination and growth [52], and, in this study, salt stress (250 mM of NaCl) impaired growth and negatively decreased root tissue length in sorghum seedlings. Carbon monoxide alleviated the negative effects of salt stress on sorghum root seedlings (Figure 1B), with an optimal concentration of 1 μM of CO. This effect might be related to the role of CO in regulating cell division and tissue proliferation [53]. Growth improvement mediated by CO has been reported in other plant species, including *Oryza sativa* [43], *Cassia obtusifolia* L. [33], and *Brassica nigra* [51], under salt and nano-silver stress.

Although there is no clear understanding of how plant cells eliminate CO, it is believed that the strong binding of CO to iron in hemoglobin is the primary inactivation method [54]. In salt stressed sorghum seedlings, Hb scavenged CO decreasing the germination index and root length (Figure 1B,D) but ZnPPIX had no effect (Figure 1B,D).

Salt stress disturbs the equilibrium balance between ROS production and scavenging, resulting in increased ROS levels [55]. The increased accumulation of ROS (H_2_O_2_) during stress is an indicator of oxidative damage to proteins, membrane lipids, and nucleic acids in plants [56,57]. Salt stress increased H_2_O_2_ content in sorghum seedlings whereas CO decreased the accumulation of H_2_O_2_ level preventing ROS-induced damage (Figure 2B). The CO scavenger (Hb) and HO inhibitor (ZnPPIX) led to increases in the H_2_O_2_ content by blocking the ameliorating effect of CO on membrane damage under salt stress (Figure 2B). These results suggest the involvement of CO in reducing ROS-induced membrane lipid peroxidation under salt stress, resulting in increased salt tolerance. The positive role of CO has been shown to reduce oxidative damage in mustard [51], cucumber [58], and wheat [45].

Similar trends of increased proline content have been observed in sorghum [6,12,14,59], *Lepidium draba* from the *Brassicaceae* family [60], and transgenic Arabidopsis plants [61] under salt stress. Proline accumulation also acts as a low-molecular-weight antioxidant that scavenges ROS, maintains membrane integrity, and stabilizes enzymes [62,63]. A further increase in proline content was observed in the presence of CO (1 and 1.5 μM of CO) under salt stress compared to seedlings treated with salt only (Figure 3B). Pretreatment with hemin under salt stress conditions has been reported to enhance proline accumulation in wild-type Arabidopsis leaves [61]. The increase in proline content induced by exogenous application of CO is linked to the activation of pyrroline-5-carboxylate synthase (P5CS) and a decrease in proline dehydrogenase (ProDH) activity, thereby alleviating salt-induced damage [64]. The exogenous application of the CO scavenger (Hb) and HO inhibitor (ZnPPIX) significantly inhibited and scavenged CO, decreasing proline accumulation. Salt stress upregulated the *SbP5CS1* transcript, which correlated with increased proline content compared to control levels in sorghum seedlings (Figure 3). These results indicate that CO plays a significant role in osmoregulation under salt stress conditions.

Vascular bundles (xylem and phloem) facilitate the transportation of water, minerals, and nutrients from roots to other plant parts [65]. The SEM micrographs revealed morphological shrinkage, and deformities of the xylem and phloem, and enlarged porous pit sizes in salt-stressed sorghum roots (Figure 4). A high salt concentration alters the porosity and hydraulic conductivity, which leads to a low water potential and eventually leads to physiological water deficit conditions, destabilization of the cell membrane, and protein degradation owing to the toxic effects of accumulated salt ions (mainly Na^+^) [66]. The shrinkage, deformities, and porosity of the vascular bundle and pit caused by the increased salt stress were reversed by CO (Figure 4). The hydrogen bonds between the water molecules generate surface tension and cause water to adhere to the hydrophilic cell wall. This, in turn, guarantees uninterrupted water transport in the xylem tissue. The effect of salt stress on xylem function is an area of active research, and understanding the role of pits in this process could provide valuable insights for developing strategies to mitigate the effects of salt stress in plants [67].

High salt concentrations cause ionic imbalance due to increased amounts of Na^+^ and Cl^−^ in the intracellular compartments, which prevents other nutrient ions from being absorbed and leads to a decrease or deficiency of essential ions, such as K^+^, Ca^+^, and Mg^2+^ [30,68]. In this study, salt stress caused increased accumulation of Na^+^ which affected the absorption of K^+^ by disrupting the Na^+^ and K^+^ balance shown by high Na^+^/K^+^ ratios of 2.06 (Figure 5A–D). CO treatment decreased Na^+^ content while K^+^ content increased, resulting in a low Na^+^/K^+^ ratio of 1.46 (1 μM of CO) and 1.27 (1.5 μM of CO) under salt stress (Figure 5C,D). This decrease in Na^+^ content might be associated with CO-induced upregulation of antiporter and transporter genes involved in potassium uptake [69]. The increase in K^+^ is key since it is the most abundant cation in plants and is important for plant processes such as protein synthesis, enzyme activation, cation–anion balance in the cytosol, and vacuoles for salt tolerance [70,71].

The transcript levels of *SbNHX4* vacuolar antiporters and the *SbKT1* transporter genes responsible for ion balance [72,73] were constitutively expressed and upregulated under salt stress (Figure 5E). Antiporter NHXs play a role in Na^+^ exclusion and vacuolar compartmentalization, and HKT also transports excess Na^+^ from the xylem to parenchyma cells to reduce salt in the shoot and maintain K^+^ balance in plants [27,74]. The *SbNHX4* and *SbKT1* transporters appear to play significant roles in Na^+^ and K^+^ homeostasis in sorghum seedlings under salt stress by enhancing the activity of internal HO. The upregulation of transporter genes under salt stress has been reported in pepper [75], barley [76], and cotton [77]. The decrease in the *SbNHX4* and *SbKT1* transcript levels in 1 μM of CO treated sorghum seedlings may be because the CO was able to facilitate the exclusion, transportation, or distribution of Na^+^ and K^+^ from sorghum seedling tissues [78,79]. The decrease coincided with element distribution in the vascular bundle indicating that CO facilitated the exclusion of Na^+^ and absorption of K^+^, returning ionic homeostasis to the sorghum seedlings under salt stress (Figure 5E).

The exposure of plants to salt results in the accumulation, changes, or damage of biomolecules including proteins, carbohydrates, lipids, and nucleic acids [80]. The FTIR spectra (Figure 6) demonstrated that salt stress caused biomolecule damage and that hematin had a protective effect, reducing salt stress.

Under stress, plants develop adaptive antioxidant mechanisms to counteract the oxidative damage caused by salt stress in the form of enzymes, such as APX, GPOX, CAT, SOD, and HO, which scavenge ROS and protect the cell [36,41,72]. This study showed that HO activity was induced under salt stress (Figure 7). The HO enzyme activity results were consistent with those of previous studies on *Amaranthus tricolor* under salt stress [73] and *Triticum aestivum* under different abiotic stresses [81]. HO1 antioxidant activity was further increased in the presence of 1 μM of CO, whereas Hemoglobin and ZnPPIX reversed this increase in sorghum seedlings. Thus, CO appears to have a cytoprotective role by increasing antioxidant activity (Figure 7A). The protective role of CO under salt stress is due to the increased level of HO1 expression, which is known to protect plants from stress-induced damage [36,82]. A downregulation was observed in the transcript level of *SbHO1* under salt stress in the presence of Hemoglobin, and the HO inhibitor (ZnPPIX), which might be due to internal CO conferring cytoprotection to sorghum seedlings [41].

The expression levels of *SbMnSOD* and *SbCAT* transcripts in the sorghum seedlings were downregulated perhaps as a result of suppression of antioxidant defense gene expression caused by increased salt levels, as reported in strawberry plants [83]. However, this study showed that 1 and 1.5 μM of CO induced the upregulation of the *SbFeSOD* and *SbMnSOD* transcripts, thereby suggesting a cytoprotective role in scavenging ROS accumulation. Under stress, SOD plays an important role in the conversion of superoxide anions to H_2_O_2_ [84] which are then scavenged by CAT into water, which might explain the observed downregulation by *SbCAT*.

These findings show the effect of salt stress, which resulted in oxidative damage causing reduced growth to sorghum seedlings. The application of CO-induced salt stress tolerance to sorghum seedlings through the synthesis of osmoprotectants, the activation of the *SbHO1* enzyme, and protein expression (Figure 10). CO improved salt tolerance by regulating ion homeostasis and antioxidant gene expressions. Treatment with the inhibitor (ZnPPIX) and scavenger (Hb) hindered the protective role of CO (Figure 10). A positive correlation was observed for all traits studied including growth parameters (germination index and root length), oxidative stress marker, H_2_O_2_, proline and *SbP5CS1* transcript, antiporter genes (*SbKT1* and *SbNHX4*), and antioxidant genes (*SbFeSOD* and *SbMnSOD*). A negative correlation was observed between growth attributes (the germination index and root length) and biochemical traits (H_2_O_2_ and proline). Proline displays positive correlations with oxidative markers and other parameters like TSS and pit diameter, indicating their potential role in mitigating oxidative stress and influencing anatomical changes.

## 4. Materials and Methods

### 4.1. Seed Preparation and Growth Conditions

Sorghum seedlings (AgFlash/NIAGARA III: (Sorghum × Sudan) and red seeds) were obtained from Agricol, Brackenfell, Cape Town, South Africa, and germinated as previously described [6]. The seeds were disinfected by incubating in 70% ethanol for 1 min with shaking at 600 rpm and rinsed three times with autoclaved distilled water. The seeds were further decontaminated by soaking in 5% sodium hypochlorite solution for 1 h with shaking at 600 rpm, followed by extensive rinsing with autoclaved distilled water. Following disinfection, the seeds were imbibed overnight in autoclaved distilled water and incubated at 25 °C with shaking at 600 rpm. The seeds were dried under laminar flow air, and uniform seeds showing radicle emergence were selected and germinated in sterile Petri dishes layered with filter paper containing 4 mL of various treatment solutions.

### 4.2. Chemicals and Treatments

Hematin (Ht) (cat # H3281, Sigma-Aldrich, Saint Louis, MO, USA) was used as the CO donor (10 mM dissolved in 1 M of NaOH), zinc protoporphyrin IX (ZnPPIX) (cat # 282820 Sigma-Aldrich, Saint Louis, MO, USA) was used as a HO1 inhibitor [40 mM dissolved in 25 mg/mL dimethyl sulfoxide (DMSO)], and hemoglobin (Hb) (cat # H4131 Sigma-Aldrich, Saint Louis, MO, USA) was used as a CO scavenger 2 g/L; dissolved in distilled water [50]. The seeds were germinated in Petri dishes containing different salt concentrations of 200 mM of NaCl (Appendix A) and 250 mM of NaCl in the absence and presence of CO (1 μM and 1.5 of Hematin), ZnPPIX (5 μM and 10 μM), and Hb (0.1 g/L). Only the 250 mM NaCl stress data are presented in the main article based on the elicited salt stress response. Each Petri dish contained five sorghum seeds and triplicate dishes were tested in each treatment. The seeds were allowed to germinate at 25 °C for 7 days in the dark and were monitored daily for growth. The root lengths of all seedlings were measured on day 7. The whole seedlings were then harvested (day 7), rinsed with distilled water to remove traces of the treatment solutions, and either used immediately or stored at −80 °C until further analysis.

### 4.3. Growth Analysis

#### 4.3.1. Germination Index

The germination index (GI) was estimated using Equation (1), which is the number of germinated seeds on the first, second, and subsequent days until the 7th day; 6, 5 …, and 1 are the weights assigned to the number of germinated seeds on the first, second, and subsequent days, respectively [85].
GI = Σ(7 × n1) + (6 × n2) +…+ (1 × n7) where n1, n2, … n7(1)

#### 4.3.2. Root Length

Root length was measured to the nearest mm using a ruler seven days after germination.

### 4.4. Histochemical Staining

The localization of H_2_O_2_ was detected as previously described [86] with minor modifications. Untreated and treated root samples of the sorghum seedlings were incubated in 1 mg/mL of 3′3′-diaminobenzidine (DAB) prepared in HCl acidified water (pH 3.8) at 25 °C. After 12 h of incubation, the roots were boiled in 90% ethanol for 15 min to reveal the reddish-brown color produced by the reaction between H_2_O_2_ and DAB.

### 4.5. Hydrogen Peroxide Content

The hydrogen peroxide (H_2_O_2_) content was measured to determine the oxidative damage caused to the sorghum seedlings using a colorimetric technique [87] with minor modifications. About 0.1 g of ground whole plant material was homogenized in 1 mL of reaction solution [0.25 mL of 0.1% (*w*/*v*) trichloroacetic acid (TCA), 0.5 mL of 1 M potassium iodide (KI), and 0.25 mL and 10 mM of potassium phosphate buffer (pH 6.8)] for 10 min at 4 °C. The homogenized samples were vortexed and centrifuged at 12,000× *g* for 15 min at 4 °C. Approximately 200 μL of supernatant from each sample was transferred to a 96 well microtiter plate and incubated for 1 h at room temperature. The absorbance was measured at 390 nm using a FLUOstar Omega microtiter plate reader (BMG LABTECH, Ortenberg, Germany). A standard curve was generated using the H_2_O_2_ solution and used to calculate the H_2_O_2_ content.

### 4.6. Proline Content

The proline content was estimated based on a previously described method [88], with a few modifications. About 0.1 g of ground whole plant material was homogenized in 500 μL of 3% (*w*/*v*) sulfosalicylic acid and centrifuged at 13,000× *g* for 20 min at 4 °C. The supernatant (300 μL) was transferred to a clean tube containing 600 μL of the reaction mixture [1% (*w*/*v*) ninhydrin dissolved in 60% (*v*/*v*) glacial acetic acid and 20% (*v*/*v*) ethanol] and incubated in a water bath at 95 °C for 20 min. The samples were allowed to cool on ice for 10 min and centrifuged at 13,000× *g* for 5 min. The absorbance was read at 520 nm using a FLUOstar^®^ Omega microtiter plate reader (BMG LABTECH, Ortenberg, Germany). The proline content was determined from a standard curve using pure proline as the standard.

### 4.7. Scanning Electron Microscopy Analysis

The anatomic structure and element distribution analyses were conducted to determine the effect of salt (250 mM of NaCl) and carbon monoxide treatment (1 μM and 1.5 μM of Ht) on sorghum roots. The analysis was conducted using a Scanning electron microscopy–energy dispersive X-ray spectroscopy (SEM-EDX) at the Physics Department, University of the Western Cape. About 0.1 g of samples of dried ground root whole plant material was placed on aluminum stubs coated with conductive carbon tape and coated with a thin layer of carbon using an EMITECH-K950x carbon coater. All EDX spectra were collected using an Oxford X-Max (Oxford Link-ISIS 300, Concord, MA, USA) silicon solid-state drift detector at an accelerating voltage of 20 kV for 60 s to ensure accurate X-ray detection. All spectra were analyzed using the Oxford Aztec software suite (Oxford Instruments plc, Abingdon, UK). The samples were then imaged, and images were collected using a Zeiss Auriga field-emission gun scanning electron microscope (Zeiss Auriga HR-SEM; Carl Zeiss Microscopy GmbH, Jena, Germany) operated at an accelerating voltage of 5 kV using an in-lens secondary electron detector [6,11].

### 4.8. FTIR Spectroscopic Analysis

Fourier-transform infrared spectroscopy (FTIR) analysis was performed to identify the types of chemical bonds (functional groups) present in the untreated and treated NaCl-stressed seedlings. Approximately 10 mg of ground and dried whole plant material was encapsulated in 100 mg of KBr pellets to prepare translucent sample discs. The powdered sample of each plant material was analyzed using a PerkinElmer Spectrum 100-FTIR spectrophotometer (PerkinElmer (Pty) Ltd., Midrand, South Africa) with a scan range from 400 to 4000 cm^−1^ and a resolution of 4 cm^−1^ [89].

### 4.9. Heme Oxygenase Activity Assay

The plant material (0.1 g) was homogenized in 3 mL of 50 mM potassium phosphate buffer (pH 7.8 containing 0.5 mM ethylenediamine tetra-acetic acid (EDTA)). The homogenate was centrifuged at 15,000× *g* for 20 min at 4 °C. Heme oxygenase activity in the supernatant was measured as previously described [90], with slight modifications. About 0.1 g of ground sorghum seedlings was homogenized in 1.2 mL of ice-cold reaction solution [prepared to contain 0.25 M sucrose solution containing 1 mM phenylmethylsulfonyl fluoride (PMSF), 0.2 mM EDTA, and 50 mM potassium phosphate buffer (pH 7.4)]. Homogenates were centrifuged at 15,000× *g* for 25 min and supernatant fractions were used as the enzyme extract for measuring the HO1 activity. The reaction mixture, in a final volume of 1 mL, contained 250 μL of enzyme extract, 200 mM hemin (cat # H9039 Sigma-Aldrich, USA), and 10 mM of potassium phosphate buffer (pH 7.4). The reaction was initiated by adding 60 nmol of NADPH, followed by incubation at 37 °C for 60 min. The concentration of biliverdin (BV) was estimated using a molar absorption coefficient at 650 nm of 6.25 mM^−1^ cm^−1^ in 0.1 M HEPES–NaOH buffer pH 7.2. One unit of HO was defined as the amount of enzyme that formed 1 nmol BV per 30 min under the assay conditions.

### 4.10. Protein Extraction and Quantification

Protein was extracted from 7 days old sorghum seedlings untreated (control) and treated (salt stressed) using the trichloroacetic acid/acetone precipitation method, as previously described [91]. About 0.2 g of ground plant material (whole seedlings) was washed with 10% TCA/acetone by vortex-mixing and centrifuging at 16,000× *g* for 3 min at 4 °C. The supernatant was decanted, and the pellet was treated with 80% methanol containing 0.1 M ammonium acetate. The mixture was then centrifuged at 16,000× *g* for 3 min at 4 °C and the supernatant was discarded. The pellet was washed with 80% acetone by vortex-mixing until fully dispersed, followed by centrifugation at 16,000× *g* for 3 min at 4 °C. The supernatant was discarded, and the pellet was air-dried at room temperature to remove any residual acetone. About 1.6 mL/0.2 g starting material of 1:1 phenol (pH 8.0, Sigma-Aldrich, Saint Louis, MO, USA)/SDS buffer [30% sucrose, 2% SDS, 0.1 M Tris-HCl pH 8, 5% mercaptoethanol] was added, mixed thoroughly, and incubated for 5 min. Following incubation, the mixture was centrifuged at 16,000× *g* for 3 min at 4 °C. The upper phenol phase was transferred to a clean tube filled with 80% methanol containing 0.1 M of ammonium acetate and stored overnight at 4 °C to precipitate the protein. The phenol phase incubated overnight was centrifuged at 16,000× *g* for 5 min at 4 °C, and the supernatant was discarded. The pellet was washed with 100% methanol and 80% acetone, vortexed, centrifuged, and the supernatant discarded. The extracted protein pellets were air-dried and resuspended in urea buffer [7 M urea, 2 M thiourea, and 4% 3, 3-chlolamidopropyl dimethylammonio-1-propanesulfomate (CHAPS)] with vigorous vortex-mixing at room temperature. The extracted protein was quantified using the Bradford assay, BSA as a standard [92].

### 4.11. Western Blot Analysis for Heme Oxygenase 

Protein extracts from Section 4.11 were analyzed by Western blotting as previously described [93], with a few modifications. Approximately 40 μg of the protein extract from the sorghum seedlings was analyzed by 14% sodium dodecyl sulfate (SDS)-polyacrylamide gel electrophoresis. The separated proteins were transferred to a polyvinylidene difluoride (PVDF) membrane (GE Healthcare, Bio-Sciences AB, Uppsala, Sweden) using a Trans-Blot^®^ Turbo Transfer system (Serial number 690BR3410, Bio-Rad, Hercules, CA, USA) for 20 min. After protein transfer, the membrane was rinsed once in 1X phosphate-buffered saline containing 0.1% Tween 20 (PBS-T) and incubated in blocking solution (1% *w*/*v*) casein from bovine milk dissolved in 1X PBS-T buffer for 1 h. The membrane was rinsed twice with PBS-T buffer and incubated overnight with primary antibody [rabbit monoclonal (EPR1390Y) to heme oxygenase 1 (Abcam, ab68477, Cambridge, UK) diluted 1:1000 in 1X PBS-T buffer. The membrane was then washed thrice with PBS-T buffer for 10 min per wash. Following this, the membrane was incubated for 1 h with the secondary antibody [goat anti-rabbit IgG StarBright™ 520 (cat # 12005870 Bio-Rad, Hercules, CA, USA)] diluted to 1:1000 with 1X PBS-T buffer. The membrane was washed thrice with PBS-T for 10 min each. The heme oxygenase-1 proteins were visualized using a ChemiDoc MP imaging system (Bio-Rad). A HSP70 monoclonal antibody as a positive control was included in the study [mouse monoclonal to HSP70 (Abcam, ab2787; Cambridge, UK)] and a secondary antibody [goat anti-mouse IgG StarBright™ blue 520 cat # 12005867 Bio-Rad, Hercules, CA, USA].

### 4.12. Total RNA Extraction and Reverse Transcription

Total RNA was extracted from 7-day-old sorghum seedlings using the Favorgen Plant Mini-RNA Extraction Kit (FAPRK 001-1 Favorgen Biotech Corp., Ping-Tung, Taiwan) according to the manufacturer’s instructions. To remove the genomic DNA, the extracted RNA was treated with an RNase-free DNase set (New England Biolabs, MA, USA) and analyzed on a 1% agarose gel. Approximately 1 μg of the total extracted RNA was used for the synthesis of first-strand cDNA using the SuperScript™ III First-Strand synthesis kit (Invitrogen, Carlsbad, CA, USA) according to the manufacturer’s instructions. The synthesized cDNA was diluted 10-fold and stored at −20 °C for semi-quantitative PCR analysis.

### 4.13. Semi-Quantitative RT-PCR

Semi-quantitative RT-PCR was used to analyze gene expression levels in sorghum under salt stress, as previously described [94] with a few modifications. Following cDNA synthesis, the PCR reaction mix contained 1 μL of a 10-fold diluted template cDNA, 12.5 μL 2X Red ampliqon Master mix (Cat # A180301, Ampliqon, Denmark), 0.2 μM of forward and reverse primer (10 μM) and distilled H_2_O added to a final volume of 25 μL. The reactions were performed at 95 °C for 3 min, 95 °C for 30 s, 30 cycles at 55 °C (*SbMnSOD*, *SbKT1*, *SbCAT*, *SbNHX4*, *18S*, and *UBQ*), 50 °C (*SbFeSOD*), and 58 °C (*SbHO1*), and 35 cycles at 60 °C (*SbP5CS1*) for 40 s, 72 °C for 30 s, and 72 °C for 5 min. Primer data for the target genes and the reference genes 18S ribosomal RNA (*18S rRNA*) used for PCR are shown in Table 1. The transcript levels of the genes were analyzed on a 1% agarose gel and normalized to the reference genes using the ImageJ data analysis software.

### 4.14. Statistical Analysis

All experiments were repeated at least four times, and the data were statistically analyzed by a one-way ANOVA using Minitab^®^ Statistical Software (http://www.minitab.com/en-us/support/downloads/, accessed on 22 November 2022). The data in the figures and tables represent the mean ± standard deviation. Statistical significance between the control and treated plants was determined using the Tukey HSD post hoc test at a 95% confidence interval and represented as *p* ≤ 0.05. Means that do not share a letter are statistically significant. Gally and “my norm” from R software (R-4.3.3) was used for Pearson’s correlation (r) matrix analysis. Appendix A for all analyses has been supplied: Effect of CO on the germination index and root length of sorghum under 200 mM salt stress (Appendix A); Effect of CO on oxidative damage to biomolecules in sorghum under 200 mM salt stress (Appendix A); Element distribution of sorghum seedlings treated with CO under salt stress (Appendix A); *SbHSP70* protein expression (Appendix A).

## 5. Conclusions

Salt-induced oxidative damage of sorghum seedlings was alleviated by hematin, a CO donor, enabling the maintenance of ion homeostasis and preventing damage to biomolecules. Additional studies on the endogenous production and role of CO in plant stress tolerance in field studies are needed to fully elucidate the role of hematin in improving salt stress tolerance in sorghum seedlings. This may lead to improved stand establishment in field settings and may persist in the alleviation of as the crop develops.

## Figures and Tables

**Figure 1 plants-13-00782-f001:**
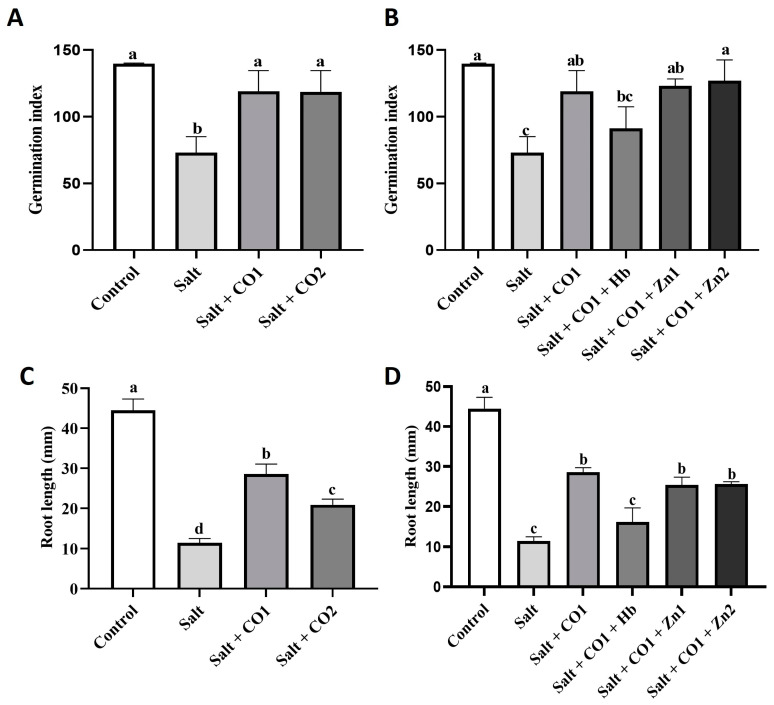
Effect of CO on germination index and root length of sorghum under salt stress. Germination index (**A**,**B**), root length (**C**,**D**) of sorghum seedlings treated with 0 mM of NaCl (control), 250 mM of NaCl (salt), a 1 μM of CO donor (Hematin; CO1), and 1.5 μM of CO donor (CO_2_), 0.1 g/L of hemoglobin (Hb), 5 μM of ZnPPIX (Zn1), and 10 of μM ZnPPIX (Zn2). The means ± SD (*n* = 3) were calculated from three biological replicates, and significant differences (*p* ≤ 0.05) were determined using ANOVA and the Tukey HSD post hoc test at a 95% confidence interval. Bars with the same letter are not significantly different.

**Figure 2 plants-13-00782-f002:**
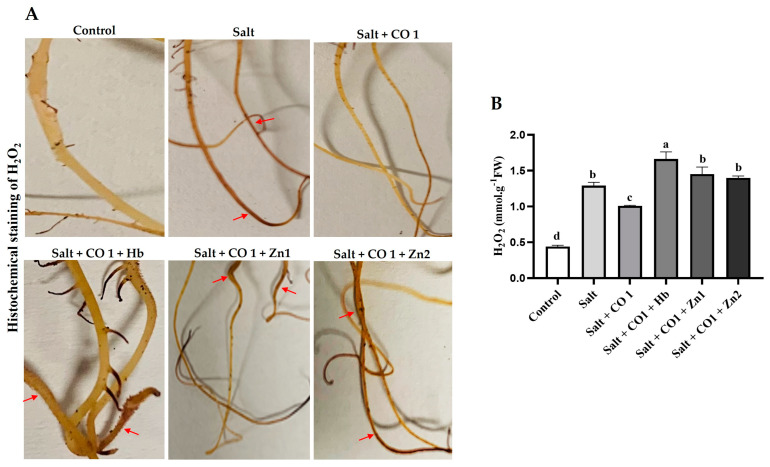
Effect of CO on the oxidative stress markers in Sorghum under salt stress. Histochemical detection of H_2_O_2_ (**A**), quantification of H_2_O_2_ (**B**) in sorghum seedlings. The red arrows on the histogram highlight the deep brown color indicative of H_2_O_2_ formation. Treatments: 0 mM of NaCl (control), 250 mM of NaCl (Salt), 1 μM of CO donor (Hematin; CO1), 0.1 g/L of hemoglobin (Hb), 5 μM of ZnPPIX (Zn1), and 10 μM of ZnPPIX (Zn2). The mean (±SD) was calculated from three biological replicates, and a significant difference (*p* ≤ 0.05) was determined using ANOVA and the Tukey HSD post hoc test at a 95% confidence interval. Bars with the same letter are not significantly different.

**Figure 3 plants-13-00782-f003:**
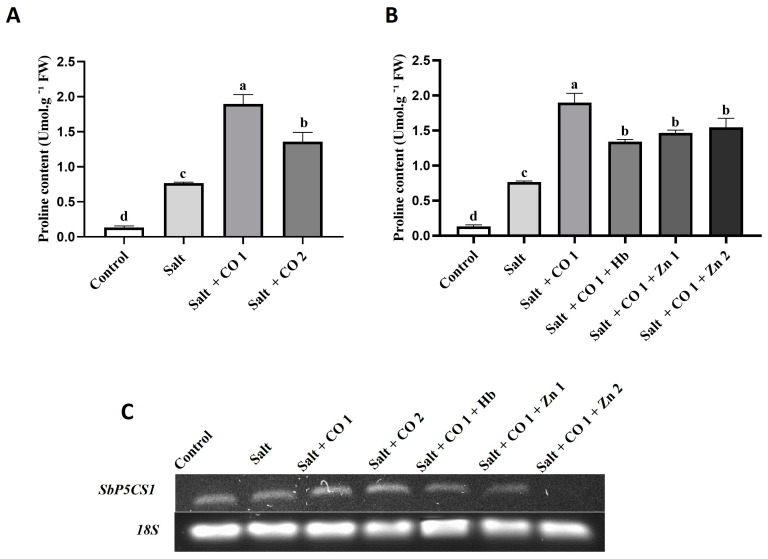
Effect of CO on osmolyte accumulation in sorghum seedlings under salt stress. Proline content (**A**,**B**), and *SbP5CS1* transcript (**C**), of sorghum seedlings. Treatments: 0 mM of NaCl (control), 250 mM of NaCl (salt), 1 μM of CO donor (Hematin; CO1), and 1.5 μM of CO donor (CO2), 0.1 g/L of hemoglobin (Hb), 5 μM of ZnPPIX (Zn1), and 10 μM of ZnPPIX (Zn2). The mean (±SD) was calculated from three biological replicates and a significant difference (*p* ≤ 0.05) was determined using ANOVA and the Tukey HSD post hoc test at a 95% confidence interval. Bars with the same letter are not significantly different. The 18s rRNA was used as a constitutive reference control.

**Figure 4 plants-13-00782-f004:**
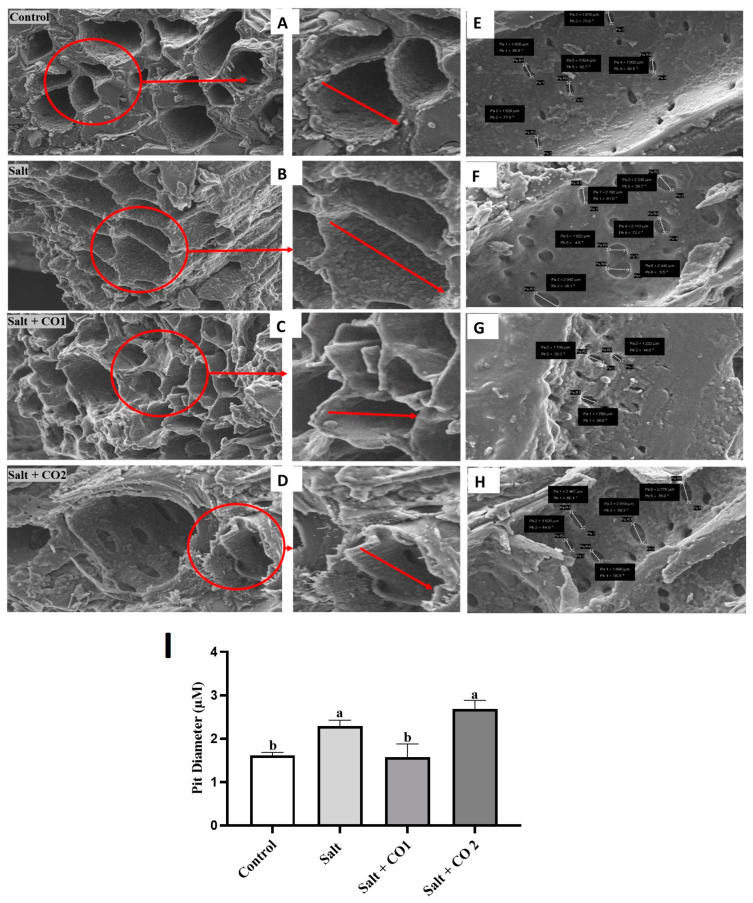
CO improves vascular bundle layers and pit sizes of sorghum under salt stress. Vascular bundle layers of sorghum roots (**A**–**D**), SEM mapping of the surface area of sorghum roots showing the xylem pits (**E**–**H**), the Xylem pit diameter of sorghum roots magnified at 2 μM (**I**). The 0 mM NaCl only (**A**,**E**), salt (**B**,**F**), salt + CO1 (**C**,**G**), salt + CO2 (**D**,**H**). Treatments: 0 mM of NaCl (control), 250 mM of NaCl (Salt), 1 μM of CO donor (Hematin; CO1), and 1.5 μM of CO donor (CO2), 0.1 g/L of hemoglobin (Hb), 5 μM of ZnPPIX (Zn1), and 10 μM of ZnPPIX (Zn2). For clarity purposes a small area showing the xylem walls in the SEM micrographs has been selected in red and enlarged/zoomed (this area is shown by red arrows). The mean (±SD) was calculated from three biological replicate pit diameter values measuring three pits per vascular bundle, and a significant difference (*p* ≤ 0.05) was determined using ANOVA and the Tukey HSD post hoc test at a 95% confidence interval. Bars with the same letter are not significantly different.

**Figure 5 plants-13-00782-f005:**
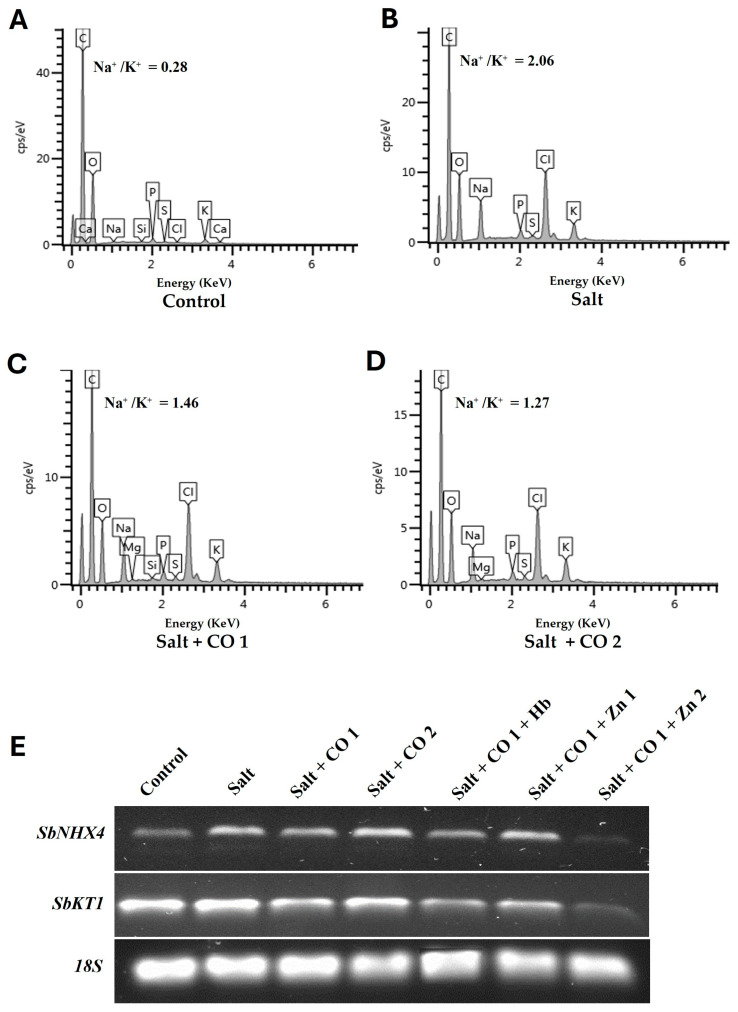
CO improved the Na^+^/K^+^ ratio and antiporter transcript levels of sorghum under salt stress. (**A**) 0 mM of NaCl only, (**B**) salt, (**C**) salt + CO1, (**D**) salt + CO2, (**E**) transcript expression levels of *SbNHX4* and *SbKT1* analyzed by semi-quantitative RT-PCR. Treatments: 0 mM of NaCl (control), 250 mM of NaCl (salt), 1 μM of CO donor (Hematin; CO1), and 1.5 μM of CO donor (CO2), 0.1 g/L of hemoglobin (Hb), 5 μM of ZnPPIX (Zn1), and 10 μM of ZnPPIX (Zn2). The 18s rRNA was used as a constitutive reference control.

**Figure 6 plants-13-00782-f006:**
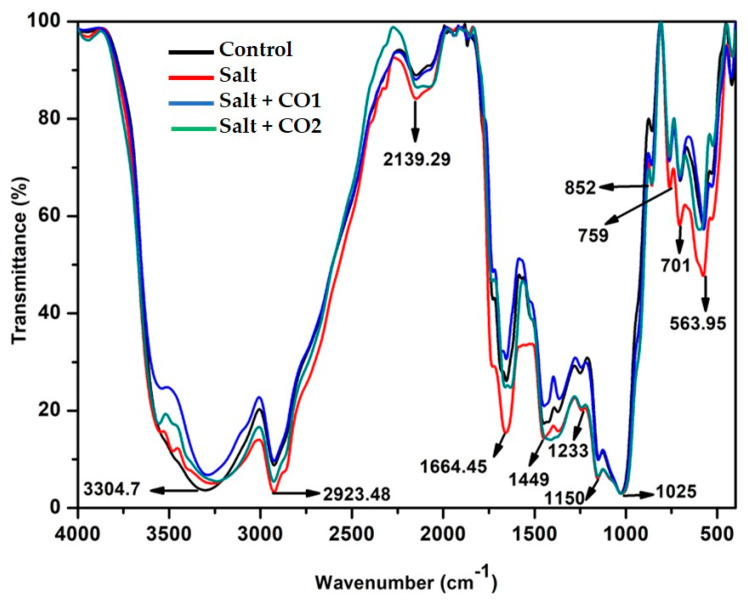
The effect of CO on FTIR spectra of sorghum biomolecules. Treatments: 0 mM of NaCl (control = black spectra), 250 mM of NaCl (salt = red spectra), 1 μM of CO donor (Hematin; CO1 = blue spectra), and 1.5 μM of CO donor (CO2 = green spectra).

**Figure 7 plants-13-00782-f007:**
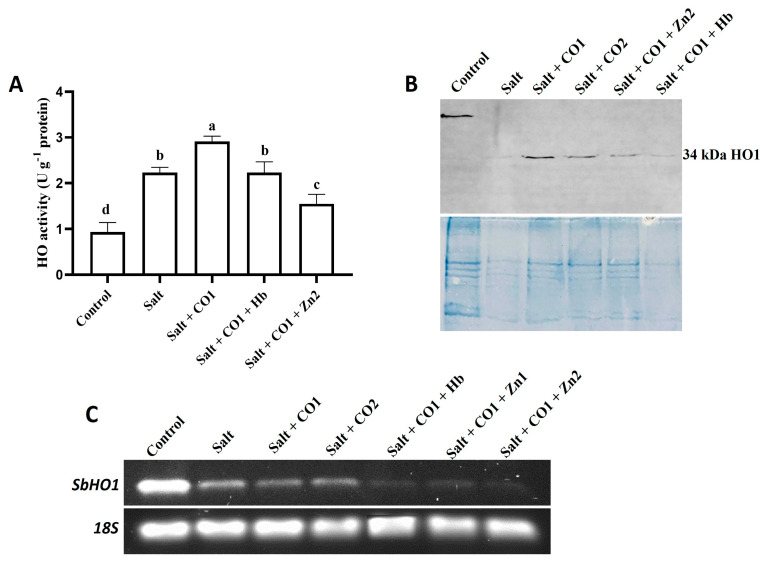
Effect of CO on enzyme activity, protein content and gene expression levels of HO in sorghum under salt stress. (**A**) Heme oxygenase activity; (**B**) Western blot analysis of the HO1 protein; lane 1: control; lane 2: salt; lane 3: salt + CO1, lane 4: salt + CO2; lane 5: salt + CO1 + Zn2; lane 6: salt + CO1 + Hb; (**C**) transcript expression level of the *SbHO1* gene analyzed by semi-quantitative RT-PCR. Treatments: 0 mM of NaCl (control), 250 mM of NaCl (salt), 1 μM of CO donor (Hematin; CO1), and 1.5 μM of CO donor (CO2), 0.1 g/L of hemoglobin (Hb), 5 μM of ZnPPIX (Zn1) and 10 μM of ZnPPIX (Zn2). The mean (±SD) was calculated from three biological replicates, and a significant difference (*p* = 0.000) was determined using ANOVA and the Tukey HSD post hoc test at a 95% confidence interval. Bars with the same letter are not significantly different. The 18s rRNA was used as a constitutive reference control.

**Figure 8 plants-13-00782-f008:**
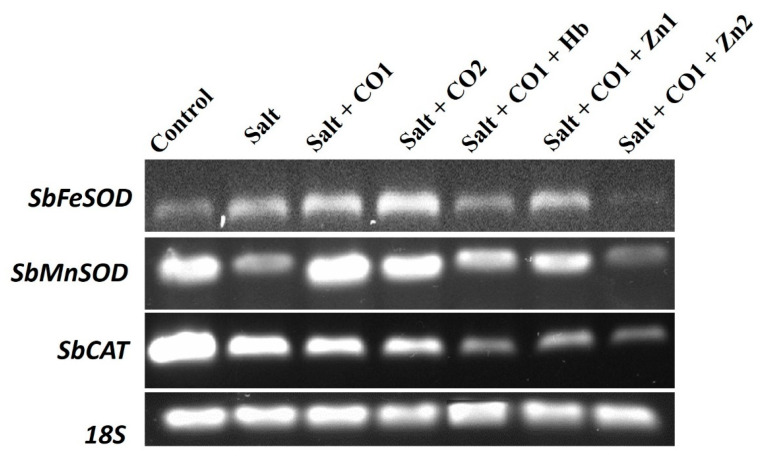
The effect of CO on the transcript levels of the *SbFeSOD*, *SbMnSOD*, and *SbCAT* genes in sorghum under salt stress analyzed by semi quantitative RT-PCR. Treatments: 0 mM of NaCl (control), 250 mM of NaCl (salt), 1 μM of CO donor (Hematin; CO1), and 1.5 μM of CO donor (CO2), 0.1 g/L of hemoglobin (Hb), 5 μM of ZnPPIX (Zn1), and 10 μM of ZnPPIX (Zn2). The 18s rRNA was used as the constitutive reference control.

**Figure 9 plants-13-00782-f009:**
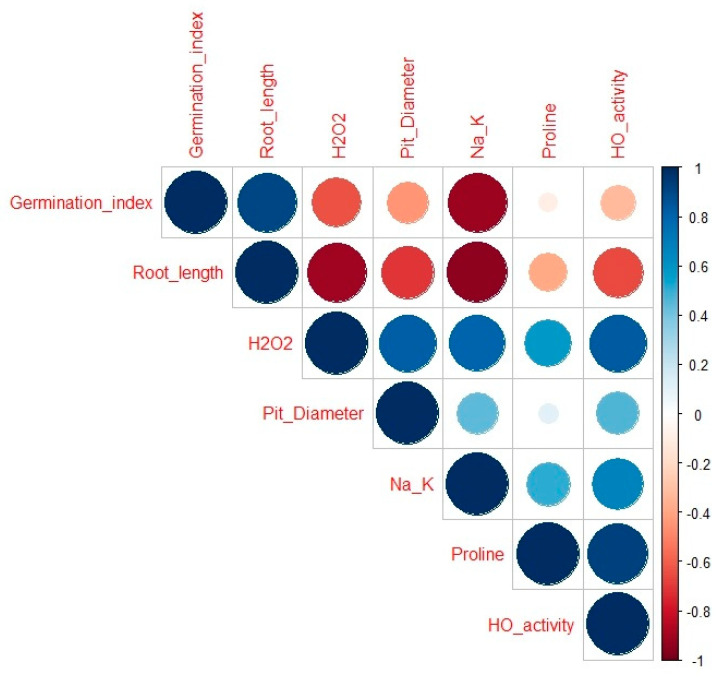
Pearson traits correlation between different parameters in response to CO under salt stress. Traits included in the analysis: germination index, root length, H_2_O_2_, proline, pit diameter, and SbHO1 enzyme activity. The color of the squares and the coefficient values indicates the strength and significance of the correlation: 0.75–1 = strongly correlated; 0.5–0.75 = highly correlated; 0.25–0.50 = moderately correlated; and 0–0.25 = weekly correlated. Under 0 = negatively correlated.

**Figure 10 plants-13-00782-f010:**
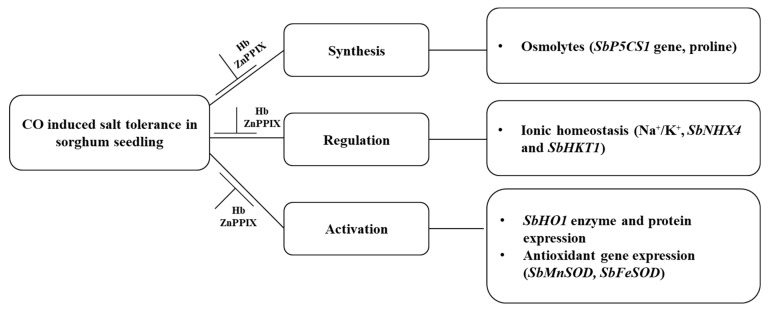
Schematic diagram of the mechanism of the role of CO in conferring salt tolerance to sorghum seedlings.

**Table 1 plants-13-00782-t001:** Gene names and accession numbers used to design primers for semi-quantitative RT-PCR analysis.

Gene Name	Forward Primer (5′-3′)	Reverse Primer (5′-3′)	Accession Number
*SbP5CS1*	CCTCTTCCCAGCTTCTTGTG	TAGCCAGAAGACCGGCTAAA	XM_021447400.1
*SbHO1*	TTCCAGACGCTCGAAGACAT	CCTGGGGATCCTTCTCAGAC	XM_002438597.2
*SbNHX4*	CATGCCACCATCATCACCAG	CTCCAAGACAATACCGCTGC	XM_021448959.1
*SbKT1*	TCCCAAAGATCAGCTGCTCA	ACGCCACTCACACAGACTTA	XM_002446325.1
*SbFeSOD*	TACGGTCTCACAACTCCACC	CAGACCTGTGCTGCATTGTT	XM_002436411.2
*SbMnSOD*	CCTTTCCCCTCCTCCATCTC	GAAGTCGTAGGAGAGGTCGG	XM_002439497.2
*SbCAT*	GGTTCGCCGTCAAGTTCTAC	AAGAAGGTGTGGAGGCTCTC	XM_021460018.1
*18S* rRNA	GCCAAGATTCAGGATAAG	TTGTAATCAGCCAATGTG	XM_002452660

## Data Availability

All data generated in this study, which are not published in this article, will be made available upon request.

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
