# Peer review of "Carbon Monoxide Alleviates Salt-Induced Oxidative Damage in Sorghum bicolor by Inducing the Expression of Proline Biosynthesis and Antioxidant Genes"

_plants, 2024, doi:10.3390/plants13060782_

Round 1

Reviewer 1 Report

Comments and Suggestions for Authors

The submitted manuscript to PLANTS-MDPI entitled “Carbon monoxide alleviates salt-induced oxidative damage in Sorghum bicolor by inducing proline biosynthesis and gene expression” is interesting to investigate. BUT, following are the comments that need to be addressed:

What were the basis to choose those studied concentrations of CO?

Please provide the research gap in the second last paragraph of introduction section.

The figures are not clear! The graph should be colorful rather than white columns to make them more attractive for the readers or at least there should be different color for different column.

The conclusion section should be improved by mentioning the responsible genes, enzymes, and proteins.

There should be a schematic diagram in the discussion section to further glorify the presentation of this hard-work.

Author Response

Dear Editor

We are grateful for the reviewers who extensively reviewed our manuscript and provided critical reports, which in turn improved our manuscript. Below are point-by-point authors responses to the comments and as implemented in the revised manuscript.

Reviewer 1

The submitted manuscript to PLANTS-MDPI entitled “Carbon monoxide alleviates salt-induced oxidative damage in Sorghum bicolor by inducing proline biosynthesis and gene expression” is interesting to investigate. BUT, following are the comments that need to be addressed:

Comment #1: What were the basis to choose those studied concentrations of CO?

Author’s Response: Initially, during the optimization stage, authors chose varying concentrations including 0.75, 1, 1.5 and 5 µM CO [which is the concentration of Hematin, the CO donor]. Results showed that the effects of 0.75 µM and 5 µM CO for most assays were not significant difference as compared to the salt stressed seedlings. [we attached some figures from the germination and root growth assays for the reviewer’s attention].

Comment #2: Please provide the research gap in the second last paragraph of introduction section.

Author’s Response: it has been improved. “Although, studies have shown the role of CO in ameliorating the effect of abiotic stresses in plants, the role of CO on a moderately salt-stress tolerant crop, sorghum remains unreported. It is therefore necessary to elucidate the role of CO in ameliorating the effect of salinity in sorghum, which will be necessary for the development of salt tolerant crops. Additionally, studies have reported on the effective use of exogenously applied phyto-protectants such as chitosan (Mulaudzi et al., 2022), calcium ion (Mulaudzi et al., 2020), methyl Jasmonate (Mulaudzi et al., 2023) and molybdenum (Mabiya et al., 2023) in mitigating the effect of salt stress in sorghum.”

Comment #3: The figures are not clear! The graph should be colorful rather than white columns to make them more attractive for the readers or at least there should be different color for different column.

Author’s Response: Figures have been re-plotted [These are attached in a PowerPoint document].

Comment #4: The conclusion section should be improved by mentioning the responsible genes, enzymes, and proteins.

Author’s Response: Authors revised and improved the conclusion.

Comment #5: There should be a schematic diagram in the discussion section to further glorify the presentation of this hard-work.

Author’s Response: Authors included the conclusion figure [Figure 10], but for we decided to include the correlation analysis [Figure 9], which brings the link between the different assays and summarizes the entire study.

Reviewer 2 comments

Comment #1: Title: Carbon monoxide alleviates salt-induced oxidative damage in Sorghum bicolor by inducing proline biosynthesis and gene expression.

Comment #2: Line 3. In the title, write fully the term “expression”. Specifying which genes is highly recommended.

Author’s Response: The title has been revised “Carbon monoxide alleviates salt-induced oxidative damage in Sorghum bicolor by inducing the expression of proline biosynthesis and antioxidant genes.”

Comment #3: Please check the use of Italics in the whole manuscript.

Author’s response: Italics have been checked.

Comment #4: The choice of the species is good and very interesting. Sorghum (Sorghum bicolor L.) is the fifth most cultivated cereal grain worldwide and is a traditional and often staple food in some arid and semi-arid regions of Africa, Central America, and Asia. I suggest to include sentences concerning this species benefits in human food and animal feed.

Author’s response: The information has been added.  “It is used as a staple food for human consumption in Africa, South Asia and Central America, and also as animal feed. Sorghum is used in the production of green fuels, including bioethanol and biogas, with low greenhouse emissions” (lines 43-46).

Comment #5: In the Introduction, authors should develop more in details the physiological, metabolic, cellular and molecular changes induced by salt in seeds, seedling and plantlets. I recommend using the following related published papers:

 Author’s response: Thank you for this comment, we revised the section and the suggested references have been implemented as shown in the sections “.  ion toxicity (Na⁺ and Cl⁻) of seed embryos (Guo et al., 2020). Salt stress had significant effect on germination percentage, germination index, shoot and root length of canola (Bybordi, 2010), basil (Çamlıca and Yaldız, 2017) and sorghum (Dehnavi et al., 2020; Mulaudzi et al., 2022). High level of salinity stress reduces physiological processes linked with decreased stomatal conductance which affects photosynthesis, chlorophyl content and transpiration (Shahid et al., 2020; Zhang et al., 2020) (lines 70 – 73).

Salinity induce osmotic and ionic imbalance mediated oxidative damage, which leads to overproduction of reactive oxygen species [hydrogen peroxide (H₂O₂), superoxide radicals (O₂⁻), hydroxyl radicals (OH), and singlet oxygen (¹O₂)] and lipid peroxidation in the cytoplasm (Hasanuzzaman et al., 2013, 2018). Increased level of reactive oxygen species (ROS) causes toxicity to the cell and disturb redox homeostasis which hinders cell division and plant growth (Arif et al., 2020; Hao et al., 2021). Oxidative stress caused by increased level of ROS in plants results in nutrient imbalances, membrane damage, inhibition of enzymatic activities, and other physiological and biochemical processes that promote plant growth or eventually lead to plant death (Tlahig et al., 2021)(Rahman et al., 2016; Li and Li, 2017; Dehnavi et al., 2020) (lines 77-82).  

1.Chemistry and Biodiversity. 2021. Response to Salinity in Legume Species: An Insight on the Effects of Salt Stress during Seed Germination and Seedling Growth. Tlahig S, Bellani L, Karmous I, Barbieri F, Loumerem M, Muccifora S. Apr;18(4):e2000917. https://doi.org/10.1002/cbdv.202000917

2.Indian Journal of Experimental Biology. 2020. Effect of salt stress on growth and physiological parameters of sorghum genotypes at an early growth stage. Fei Zhang, Suraj Sapkota, Anjan Neupane, Jialin Yu, Yanqiu Wang, Kai Zhu, Feng Lu, Ruidong Huang, Jianqiu Zou. Vol. 58, June 2020, pp.404-411.

  1. Agronomy 2020. Rajabi Dehnavi, Ahmad, Morteza Zahedi, Agnieszka Ludwiczak, Stefany Cardenas Perez, and Agnieszka Piernik. Effect of Salinity on Seed Germination and Seedling Development of Sorghum (Sorghum bicolor (L.) Moench) Genotypes" 10, no. 6: 859. https://doi.org/10.3390/agronomy10060859

Comment #6: Authors should also include references about the exogenously applied molecules, phytohormones or chemical compounds to regulate/improve plant responses to salt stress.

Author’s response: References were added. Several studies have shown the effective use of exogenously applied phyto-protectants such as chitosan (Mulaudzi et al., 2022), calcium ion (Mulaudzi et al., 2020), methyl jasmonate (Mulaudzi et al., 2023) and molybdenum (Mabiya et al., 2023) in mitigating the effect of salt stress in sorghum (lines 115-118).

Comment #7: Section 4. Materials and Methods: In paragraph 4.1. A more appropriate use instead of the term decontaminated is “disinfected”. Please correct this.

Author’s response: Decontaminated was replaced by disinfected.

Comment #8: Also, in this procedure, sodium hypochlorite is usually used prior to ethanol. Please check this protocol.

Author’s response: It is true that some protocols use sodium hypochlorite before ethanol. It is also true that some protocols use ethanol before the use of sodium hypochlorite for the disinfection step. Hence, in this research we chose the method that has been working in our research group for the past 9 years as indicated in several of our published articles [DOI 10.1007/s12192-015-0591-2; doi: 10.3390/plants9060730;] among others.

Comment #9: According to authors, each petri dish contained five sorghum seeds with varying concentrations of the treatment in three replicates. This means 5 × 3= 15 seeds/treatment. This number of biological replicates is very low to conduct physiological study. Please explain.

Author’s response: Each petri dish contained 5 seeds x 3 = 15 seeds (biological replicate 1) x 4 (repeats of independent experiments conducted at different time frames) = 60 seeds sampled. The best 3 of 4 biological replicates are chosen for final analysis and plotting the figures/graphs. See attached spreadsheet.

Comment #10: Please, mention the solvent, whether it be water or another medium, employed for the solubilization of each chemical (Hematin, Zinc protoporphyrin, Hemoglobin), and elucidate the rationale behind selecting the concentrations (along with any pertinent references or preliminary findings).

Author’s response: A stock of 10 mM Hematin was dissolved in 1 M NaOH and dilutions were prepared using distilled water. Reasons for choosing the used concentrations are fully explained under Reviewer 1’s comment #1, initially, four concentrations were chosen [0.75, 1, 1.5 and 5 µM CO], however 1 and 1.5 µM CO were chosen as the most effective concentrations [see attached figure]. Similarly, 5, 10 and 20 µM concentrations of ZnPPIX were sampled and only 5 and 10 µM were chosen based on their effective inhibition. A 40 mM ZnPPIX stock solution was prepared by dissolving 2.5 % DMSO and was made up to the final volume with autoclaved distilled water.

A stock of 2 g/L Hemoglobin was prepared in autoclaved distilled water.

For Hemoglobin, 0.1, and 0.2 g/L concentrations were used, and only 0.1 g/L Hb was best suited for further analysis (data for 0.2 g/L was not noted since 0.1 g/L was sufficient to scavenge CO production in sorghum seedling)

The concentrations for the inhibitor (ZnPPIX) and the scavenger (Hb) were also chosen based on their effectiveness in inhibiting Heme oxygenase activity and scavenging CO respectively and also based on previous research [articles shown below have been referenced].

  1. K. Liu et al., “Carbon monoxide counteracts the inhibition of seed germination and alleviates oxidative damage caused by salt stress in Oryza sativa,” Plant Sci., vol. 172, no. 3, pp. 544–555, 2007, doi: 10.1016/j.plantsci.2006.11.007.
  2. S. Xu et al., “Carbon monoxide alleviates wheat seed germination inhibition and counteracts lipid peroxidation mediated by salinity,” J. Integr. Plant Biol., vol. 48, no. 10, pp. 1168–1176, 2006, doi: 10.1111/j.1744-7909.2006.00337.x.

Comment #11:  Specify which plant material was used in the measurements of proline, H2O2, Malondialdehyde and total soluble sugar, as well as enzyme activity. Specify how many biological replicates were carried out in each experiment.

Author’s response: “whole” plant material was used for these assays. Three biological replicates were carried out in each experiment. All statistical replicates analysed are indicated in section 4.16. We also replied to the question on biological replicates under the comments given by reviewer 1 comment #9.

All experiments were repeated at least three times, and the data were statistically analyzed by a one-way ANOVA using Minitab® Statistical Software (http://www.minitab.com/en-us/support/downloads/, accessed November 2022). Data in the Figures and Tables represent the mean ± standard deviation. Statistical significance between the control and treated plants was determined using the Tukey HSD post-hoc test at a 95 % confidence interval and represented as p ≤ 0.05. Means that do not share a letter are statistically significant. Supplementary data for all analyses has been supplied; Effect of CO on the germination index and root length of S. bicolor under 200 mM salt stress (Figure S1), Effect of CO on oxidative damage to biomolecules in S. bicolor under 200 mM salt stress (Figure S2), Element distribution of sorghum seedlings treated with CO under salt stress (Table S1), SbHSP70 protein expression (Figure S3).

Comment #12: In section 4.9. Scanning electron microscopy analysis; authors should include the detailed steps of sample preparation before coating and SEM observation.

Author’s response: Samples were grinded with liquid nitrogen and dried at 80 °C overnight before sent for SEM analysis.

Comment #13: In section 4.10. FTIR Spectroscopic Analysis; There is a considerable amount of misinformation regarding the application of FTIR for identifying chemical groups in powdered samples of various plant materials, including the specific plant material in question. ICP or GC-MS would be more appropriate. Powdered sample does not allow to display functional groups on the surface but in overall sample.

Author’s response: FTIR is used to define vibrational deformities of chemical bonds.  FTIR analysis identifies functional groups (biomolecules) of liquid, gaseous, and solids through infrared absorption spectrum (it has been indicated as such in the methodology) in each sample tested.

Tkachenko, Y.; Niedzielski, P. FTIR as a Method for Qualitative Assessment of Solid Samples in Geochemical Research: A Review. Molecules 2022, 27, 8846. https://doi.org/10.3390/molecules27248846.

Comment #14: Specify which proteins in section 4.12. Protein extraction and quantification. Also add citation for Bradford assay.

Author’s response: This information has been updated. The protein extracted in section 4.12 is Heme oxygenase-1. The citation for Bradford assay has been added Bradford MM (May 1976). "A rapid and sensitive method for the quantitation of microgram quantities of protein utilizing the principle of protein-dye binding". Anal. Biochem72 (1–2): 248–54. doi:10.1016/0003-2697(76)90527-3. PMID 942051. S2CID 4359292.

Comment #14: Protein extraction was conducted with 7-days old sorghum seedlings. Specify which part of seedlings was used; root, hypocotyl, epicotyl, cotyledon, leaf?

Author’s response: We used “Whole” plant material, which were harvested on day 7, rinsed with distilled water to remove traces of the treatment solutions, and either used immediately or stored at -80 °C until further analysis (section 4.2).

Where necessary the type of plant material used for any assays has been specified such as in Section 4.4 and section 4.9.           “for Histochemical staining (Section 4.4) and “Scanning electron microscopy analysis Anatomic structure and element distribution (Section 4.9): root samples were used.

Comment #15: Authors should check and revise (personalize) the sections: Author Contributions, Funding, Data availability, Acknowledgement and Conflicts of interest.

Author’s response: These sections have been revised.

Comment #16: One of the major flaws of the article is their figures. All the figures added to the article are blurred which makes the reviewer doubt about the authenticity of the data. All the graphs should be replotted, and high-resolution figures should be added to the original article, without that the article cannot be accepted for publication.

Author’s response: All graphs have been replotted and added in the main document.

Comment #17: The quality of English should be improved.

Author’s response: Authors revised the English.

To the Editor

Below are additional changes made but not requested by the reviewers

  • The MDA (Figure 2) and total soluble sugar (Figure 3) images were removed as there was no significant difference between the control and the salt treatment.
  • Also, the percentages for the gene expression analysis were removed based on an advice to better quantify from a real time qPCR which gives qualitative data.

Reviewer 2 Report

Comments and Suggestions for Authors

Comments to authors:

Title: Carbon monoxide alleviates salt-induced oxidative damage in Sorghum bicolor by inducing proline biosynthesis and gene expression.

Line 3. In the title, write fully the term “expression”. Specifying which genes is highly recommended.

Please check the use of Italics in the whole manuscript.

The choice of the species is good and very interesting. Sorghum (Sorghum bicolor L.) is the fifth most cultivated cereal grain worldwide and is a traditional and often staple food in some arid and semi-arid regions of Africa, Central America, and Asia. I suggest to include sentences concerning this species benefits in human food and animal feed.

In the Introduction, authors should develop more in details the physiological, metabolic, cellular and molecular changes induced by salt in seeds, seedling and plantlets. I recommend using the following related published papers:

1.Chemistry and Biodiversity. 2021. Response to Salinity in Legume Species: An Insight on the Effects of Salt Stress during Seed Germination and Seedling Growth. Tlahig S, Bellani L, Karmous I, Barbieri F, Loumerem M, Muccifora S. Apr;18(4):e2000917. https://doi.org/10.1002/cbdv.202000917

2.Indian Journal of Experimental Biology. 2020. Effect of salt stress on growth and physiological parameters of sorghum genotypes at an early growth stage. Fei Zhang, Suraj Sapkota, Anjan Neupane, Jialin Yu, Yanqiu Wang, Kai Zhu, Feng Lu, Ruidong Huang, Jianqiu Zou. Vol. 58, June 2020, pp.404-411.

3. Agronomy 2020. Rajabi Dehnavi, Ahmad, Morteza Zahedi, Agnieszka Ludwiczak, Stefany Cardenas Perez, and Agnieszka Piernik. Effect of Salinity on Seed Germination and Seedling Development of Sorghum (Sorghum bicolor (L.) Moench) Genotypes" 10, no. 6: 859. https://doi.org/10.3390/agronomy10060859

Authors should also include references about the exogenously applied molecules, phytohormones or chemical compounds to regulate/improve plant responses to salt stress.

Section 4. Materials and Methods: In paragraph 4.1. A more appropriate use instead of the term decontaminated is “disinfected”. Please correct this. Also, in this procedure, sodium hypochlorite is usually used prior to ethanol. Please check this protocol.

According to authors, each petri dish contained five sorghum seeds with varying concentrations of the treatment in three replicates. This means 5 × 3= 15 seeds/treatment. This number of biological replicates is very low to conduct physiological study. Please explain.

Please, mention the solvent, whether it be water or another medium, employed for the solubilization of each chemical (Hematin, Zinc protoporphyrin, Hemoglobin), and elucidate the rationale behind selecting the concentrations (along with any pertinent references or preliminary findings).

Specify which plant material was used in the measurements of proline, H2O2, Malondialdehyde and total soluble sugar, as well as enzyme activity. Specify how many biological replicates were carried out in each experiment.

In section 4.9. Scanning electron microscopy analysis; authors should include the detailed steps of sample preparation before coating and SEM observation.

In section 4.10. FTIR Spectroscopic Analysis; There is a considerable amount of misinformation regarding the application of FTIR for identifying chemical groups in powdered samples of various plant materials, including the specific plant material in question. ICP or GC-MS would be more appropriate. Powdered sample does not allow to display functional groups on the surface but in overall sample.

Specify which proteins in section 4.12. Protein extraction and quantification. Also add citation for Bradford assay.

Protein extraction was conducted with 7-days old sorghum seedlings. Specify which part of seedlings was used; root, hypocotyl, epicotyl, cotyledon, leaf?

Authors should check and revise (personalize) the sections: Author Contributions, Funding, Data availability, Acknowledgement and Conflicts of interest.

One of the major flaws of the article is their figures. All the figures added to the article are blurred which makes the reviewer doubt about the authenticity of the data. All the graphs should be replotted, and high-resolution figures should be added to the original article, without that the article cannot be accepted for publication.

Comments on the Quality of English Language

The quality of English should be improved.

Author Response

(The authors gave the same response as above.)

Round 2

Reviewer 2 Report

Comments and Suggestions for Authors

The authors have corrected the article to a large extent adding scientific soundness to the paper. Although, I can see that the references are still messed up. Some references are written that they are added but they are not. For example: Tlahig. et al 2021. 

The references should be done using Mendley or Endnote and all the references should be added and numbered properly to avoid these mistakes, before the publication of the article.